# A Multicentre Pilot Study of a Two-Tier Newborn Sickle Cell Disease Screening Procedure with a First Tier Based on a Fully Automated MALDI-TOF MS Platform

**DOI:** 10.3390/ijns5010010

**Published:** 2019-01-23

**Authors:** Pierre Naubourg, Marven El Osta, David Rageot, Olivier Grunewald, Gilles Renom, Patrick Ducoroy, Jean-Marc Périni

**Affiliations:** 1Biomaneo, 22B boulevard Winston Churchill, F-21000 Dijon, France; 2CLIPP, Clinical Innovation Proteomic Platform, Université de Bourgogne Franche Comté, F-21000 Dijon, France; 3Newborn Screening Laboratory, Biology and Pathology Center, Lille University Medical Centre, F-59000 Lille, France

**Keywords:** newborn screening, sickle cell disease, MALDI-TOF, mass spectrometry, thalassemia, prevention

## Abstract

The reference methods used for sickle cell disease (SCD) screening usually include two analytical steps: a first tier for differentiating haemoglobin S (HbS) heterozygotes, HbS homozygotes and β-thalassemia from other samples, and a confirmatory second tier. Here, we evaluated a first-tier approach based on a fully automated matrix-assisted laser desorption/ionization time-of-flight mass spectrometry (MALDI-TOF MS) platform with automated sample processing, a laboratory information management system and NeoSickle^®^ software for automatic data interpretation. A total of 6701 samples (with high proportions of phenotypes homozygous (FS) or heterozygous (FAS) for the inherited genes for sickle haemoglobin and samples from premature newborns) were screened. The NeoSickle^®^ software correctly classified 98.8% of the samples. This specific blood sample collection was enriched in qualified difficult samples (premature newborns, FAS samples, late and very late samples, etc.). In this study, the sensitivity of FS sample detection was found to be 100% on the Lille MS facility and 99% on the Dijon MS facility, and the specificity of FS sample detection was found to be 100% on both MS facilities. The MALDI-MS platform appears to be a robust solution for first-tier use to detect the HbS variant: it is reproducible and sensitive, it has the power to analyze 600–1000 samples per day and it can reduce the unit cost of testing thanks to maximal automation, minimal intervention by the medical team and good overall practicability. The MALDI-MS approach meets today’s criteria for the large-scale, cost-effective screening of newborns, children and adults.

## 1. Introduction

The use of mass spectrometry (MS) to screen newborns for haemoglobinopathies differs from the MS-based diagnosis of a broad range of clinically significant haemoglobin (Hb) variants. Newborn screening requires a robust, high-throughput and cost-effective MS method that solely detects the biomarkers of sickle cell disease (SCD) and β-thalassemia major. In contrast, the broader diagnostic procedure involves the detection of as many changes in globin proteins as possible in a single step. Newborn screening programs are typically organised as single-tier or two-tier procedures. In the two-tier procedure, the first tier is typically a routine MS method (electrospray ionization (ESI)-MS [1,2,3] or matrix-assisted laser desorption/ionization (MALDI)-MS [4,5] capable of analysing intact globin chains. Pathological samples with a single mutation in the Hb β-chain can thus be unambiguously classified into three groups: heterozygotes without HbS variants, heterozygotes with HbS variants and HbS homozygotes [5]. Furthermore, β-globin production defects (β-thalassemia) can be detected [5]. The second tier is based on standard methods (such as HPLC, isoelectric focusing, and capillary electrophoresis (CE)) in order to distinguish between heterozygotes Hb F/AS (FAS) and compound heterozygotes (F/SC, F/SD, F/SE and F/S-OArab). In a single-tier procedure, only the most common variants (e.g., HbS, HbC, D^Punjab^, OArab and HbE) are detected in a tryptic peptide analysis of haemoglobin using ESI-MS/MS [4,6,7,8] or the direct surface sampling of dried blood spots coupled to high-resolution MS [9,10,11].

In a multicentre pilot study, we evaluated a two-tier newborn SCD screening procedure in which the first tier is based on the fully automated MALDI-TOF MS classification of Hb profiles as FA, FS or FAS. To achieve this “plug-and-play” first tier, we maximised the performance levels of our previously described preanalytical and analytical procedures [5]. Likewise, we improved the software used for automatic data interpretation. Lastly, a laboratory information management system was set up to ensure data traceability. We investigated the usability of this MALDI-TOF MS newborn SCD screening platform by assessing its analytical throughput, user-friendliness, ease of implementation, potential interfering factors, MS spectral quality and percentage of correctly classified samples. We established quality criteria for standard and nonstandard MS profiles as a function of the newborn’s phenotype, determined our algorithm’s ability to correctly classify MS profiles and identified interfering factors and their causes. Lastly, we performed a preliminary analysis of the impact of β chain variants other than HbS on the MALDI-TOF profile. Newborns having received one or more blood transfusions were excluded from the present analysis, and are being assessed in a separate, dedicated study. Our results demonstrated the potential value of a MALDI-TOF MS approach for high-throughput newborn SCD screening.

## 2. Materials and Methods

### 2.1. Sample Collection

Residual blood spots from standard Guthrie cards (used in our laboratory’s routine screening activity) were investigated with MALDI-TOF MS. All samples were collected with the parents’ consent for use in the French national newborn screening programme. No additional consent was required because the specimens were not used for purposes other than for which the blood sample was initially collected. The source of the samples has been described previously [5]. The study was registered with the French National Consultative Committee on Information Processing in Medical Research (*Comité consultatif sur le traitement de l’Information en matière de recherche dans le domaine de la santé*; reference: 14.818 12/23/2014). After collection, the samples were anonymised and stored at +4 °C.

Sets of samples from our routine activity were selected each week, according to the following criteria: (i) samples presenting a β chain variant in CE and HPLC assays; (ii) samples from premature newborns (gestational age at delivery: 22–32 weeks); (iii) samples corresponding to late and very late screening; and (iv) and randomly selected samples of daily screening activity that had an interpretable CE profile.

We analyzed 6701 samples with full datasets (i.e., with clinical data as follows: N° of sample, sex, term, weight, name of maternity, transfusion, maternity location and date of screening, birth date plus validated results using CE in the first line and HPLC in the second line; Table 1). The percentage of FAS and FS phenotypes was higher for the premature newborns than for a standard population of newborns. The sample collection date was optimal (3–5 days after delivery) for the great majority of newborns.

### 2.2. Definition of Corrected Phenotypes

Subjects who were heterozygous or homozygous for HbC, HbD-Punjab, HbE, HbO-Arab, HbKorle-Bu and some unidentified variants (HbX) according to conventional methods were detected as HbA by linear MALDI-TOF due to the technique’s lack of mass resolution. Hence, these newborns were classified as having a corrected FA phenotype. Likewise, corrected FAS phenotypes included the composite heterozygous FSC, FSE and FS-OArab phenotypes.

### 2.3. Sample Processing and Analysis

In the present pilot study, all samples were measured with MALDI-MS at two analytical facilities (the University of Burgundy’s CLIPP facility (Dijon, France) and Lille University Hospital’s newborn screening laboratory (Lille, France)) after two chips (diameter: 3.5 mm) had been isolated from each blood spot.

### 2.4. Sample Preparation for MS Measurements

At both facilities, samples were prepared for MS measurements using a research version of the NeoSickle^®^ kit (Biomaneo, Dijon, France) for an EVO 200 automated system in Dijon and an EVO 100 automated system in Lille (Tecan, Lyon, France). The NeoSickle^®^ kit was used according to the manufacturer’s instructions, which included a first step of solubilization of proteins by a specific solution, and a second step of mixing with the matrix that was specially adapted for the MALDI-TOF MS analysis of blood proteins. The sample–matrix mixture was then deposited in quadruplicate on a 384-spot polished steel MALDI target (Bruker Daltonik GmbH, Bremen, Germany).

### 2.5. Mass Spectrometry Measurements

At both facilities, MS was performed with a MALDI-TOF system (an AutoFlex™ Speed with a 2000 Hz Smartbeam™ II laser (Bruker Daltonik GmbH) in Dijon, and an AutoFlex™ III with a 200 Hz Smartbeam™ laser (Bruker Daltonik GmbH) in Lille.

### 2.6. Data Processing

Mass spectrometry acquisitions were analyzed by the algorithm if (i) the whole spectrum and the region of interest were sufficiently intense, (ii) the baseline was not too noisy and (iii) at least three of the four profiles per sample could be interpreted automatically. After evaluation of the quality of each spectrum, the spectra were normalized, smoothed and underwent baseline subtraction. The four spectra obtained of the same sample were then averaged to obtain a mean profile, which was submitted to the algorithm. 

### 2.7. Analytical Data Flow

An algorithm for automatic discrimination between normal samples and samples containing an HbS variant had been developed using spectra from an initial cohort of phenotyped samples by CE and HPLC. It has since been improved as further data are acquired. All of the analytical results were centralized via a secure data collector and submitted to the algorithm. The median newborns’ profiles were automatically classified as FA, FAS or FS.

### 2.8. Visual Assessment of MS Profiles

In order to evaluate the algorithm’s ability to classify a newborn’s profile as FA, FAS or FS, all of the following profiles were visually inspected: (i) samples classified as FS or FAS by CE/HPLC, (ii) samples for which MALDI-MS and CE/HPLC gave conflicting results, (iii) samples with abnormal CE and/or HPLC profiles, (iv) samples from premature newborns, (v) samples with β chain variants other than HbS, (vi) samples with low-intensity MS signals at 15,850 *m*/*z* and 15,880 *m*/*z* [5], (vii) samples with a “low HbA” warning and (viii) samples for which the Dijon and Lille facilities gave conflicting results. Lastly, half the remaining samples (classified as FA by both HPLC and CE) were checked at random.

### 2.9. The Data Collector

Sickle Cell Anemia Collect & Compare (SCACC) is a web application developed by Biomaneo to help biologists compare MS screening results with those of the reference screening methods (CE and HPLC). Within a single application, SCACC can store, group and present heterogeneous data in a user-friendly way, including clinical data on the sample donor, the experimental data from CE and HPLC analyses, the experimental data from the MS analysis and the validation files (i.e., the screening results sent to the paediatrician).

The SCACC application contains a table in which all of the information available for a given sample is shown on a single line. Along with a filter system for each variable, this layout makes it easy to create pools of interest (preterm samples, pathologic results, etc.) for the analysis of any misinterpreted results.

## 3. Results

### 3.1. Optimization of Preanalytical and Analytical Procedures

In France, newborn SCD screening is centralized at a few specialist analytical centres. This centralisation requires a very-high-throughput system capable of analyzing 600–1000 samples per day. The MALDI-TOF MS approach meets this requirement: for each run, it takes 1 h to deposit 288 samples in quadruplicate on three 384-spot MALDI targets.

Using an AutoFlex™ Speed system, it took 22 min to analyze a MALDI target containing 96 samples. Hence, the analytical throughput was estimated to be around 210 samples/hour; it took 5 h to analyze 1056 samples.

One of the essential criteria for automation was easy operation of the software that controlled the platform and displayed the results. A technician with little experience of MS could perform analyses on the MALDI-TOF MS system interfaced with the workstation. Another strong point is that all of the data from an analytical run were published and aggregated into a 96-well plate format. Abnormal profiles were differentiated from normal profiles by a colour code. Three different icons can be showed within a given coloured square: a cross indicating that the profile could not be interpreted, an exclamation mark indicating that the profile’s initial automatic classification has been modified manually or a question mark indicating that the profile did not match the phenotype and must be checked by the operator. The spectrum’s region of interest can be displayed in a mode in which β, βS and γ chains are seen or in a zoom mode in which only the β and βS chains are seen (Figure 1). The abnormal MS profiles were assessed visually by the operator. When applicable, two decision-support warnings could be displayed by the algorithm: “low HbA concentration” or “newborn having received a blood transfusion”. The data for a given newborn (ID number, demographic data, zoom mode or not, classification changes and non-hidden alerts) can be displayed by clicking on a well.

### 3.2. Spectral Quality of MALDI-TOF Data

Taken as a whole, our results show NeoSickle^®^’s ability to correctly classify 97% of the samples tested in Lille and 98.8% of the samples tested in Dijon. The sensitivity of the NeoSickle^®^ approaches to detect the FA samples is 99.2% and 97.2% with a specificity of 99% and 99.3% on the Lille and the Dijon MS facility, respectively. The detection of an HbS chain in samples is obtained on the Lille and the Dijon MS facility with a sensitivity of 98.4% and 97% and with a specificity of 99.8% and 99%, respectively.

The positive predictive value of FS profiles detection was 96.6% and 97.7% with the results obtained by the Lille and the Dijon MS facility, respectively. The negative predictive value of FS profiles detection obtained at Lille and the Dijon MS facility was 100% and 99.9%, respectively.

It is important to note that these results were obtained with biased blood samples that were enriched in FS and FAS samples and difficult samples (premature newborns, very late screening).

#### 3.2.1. Criteria Used to Define a Standard MALDI-TOF Profile

In 85.8% of the analyses in Dijon and 92.9% of the analyses in Lille, the median FA, FAS, FS and S-β^+^ MS profiles (Figure 2a,c,e,g, respectively) were considered to be standard and were automatically interpreted by the NeoSickle^®^ software. Limits not to be exceeded were defined: an FA profile with a low-noise ascending or convex baseline was considered to be normal (Figure 2b); an FAS profile was considered to be standard as long as a plateau separated the β^S^ and β^A^ chain peaks (Figure 2d); and, lastly, an FS profile with a low-noise ascending or convex baseline to the right of the β^S^ chain peak was considered to be interpretable (Figure 2f). The profile of a newborn with S-β^+^-thalassemia was characterized by a much more intense β^S^ peak, relative to the β^A^ peak (Figure 2g,h).

#### 3.2.2. Classification by the Algorithm of “Standard” Profiles, as a Function of the Newborn’s Corrected Phenotype

There were no classification errors among the corrected FA and FS samples (Table 2). Concerning the corrected FAS samples, only 2 of the 2753 standard profiles from Lille were misclassified (as FS) by MS, and 17 of the 2507 standard profiles from Dijon were misclassified (16 cases classified as FA by MS, and 1 case classified as FS by MS). Although the spectra of the FAS samples misclassified as FA showed a shoulder with a plateau at 15,850 *m*/*z* ± 10 *m*/*z*, the signal intensity was too low. This characteristic should be introduced in the criteria for a nonstandard MS profile.

The FAS samples misclassified as FS by MALDI-MS came from composite heterozygous newborns classified as FS-O Arab and FSE by CE and HPLC. Regular peaks at 18,850 and 1880 *m*/*z* were observed; however, the error was induced by the low intensity of the peak at 15,880 *m*/*z*. A profile with asymmetry for the β^A^ and β^S^ chains (a more intense β^S^ peak) prompted its classification as FS by the e-NeoSickle software.

The 16 samples misclassified in Dijon gave good results in Lille, indicating that the errors were probably due to a temporary lack of spectrum quality and not a sample or detection problem. The quality of the Dijon data was not as high as in Lille because the number of nonstandard spectra was higher; this finding agrees with the higher misclassification rate observed in Dijon.

#### 3.2.3. Description of a Nonstandard MALDI-MS Profile

A nonstandard median MS profile was detected in 14.7% of the analyses in Dijon and 7.6% of the analyses in Lille. The MS profiles were considered to be nonstandard (Figure 3) for one of three reasons. Firstly (cause 1, see Table 3), some profiles had a nonregular baseline or a slightly distorted peak; this applied to (i) FA MS profiles with a variably broad/sharp/distorted peak but very low intensity at 15,837 *m*/*z* ± 5 Da, relative to the β^A^ peak (Figure 3a,b); an FAS MS profile with minor deformation of the β^S^ peak (Figure 3c); and FS and S-β^+^-thalassemia MS profiles with an irregular baseline (data not shown) and a deformation of the β^A^ peak (Figure 3d). Secondly (cause 2), some profiles had a low, broad peak at 15,837 *m*/*z*; this applied to (i) FA MS profiles with a variably shifted peak or (ii) FAS MS profiles characterized by a broad, well-centred peak at 15,837 ± 5 *m*/*z* but that was much less intense than the β^A^ chain peak at 15,867 *m*/*z* (data not shown). Thirdly (cause 3), some FA MS profiles had a broader β^A^ peak that variably overlapped with the region of interest at 15,837 *m*/*z* and led to misclassification as “FAS” (Figure 3e), whereas other FAS MS profiles showed a low resolution and thus poor separation of the β^A^ and β^S^ peaks, giving a single large peak as a shoulder and no plateau between the peaks (Figure 3f).

#### 3.2.4. Classification by the Algorithm of Profiles Considered to be Nonstandard in a Visual Assessment, as a Function of the Newborn’s Corrected Phenotype

The data in Table 3 show that the great majority of nonstandard profiles (84.5% in Lille and 81.2% in Dijon) were nevertheless correctly classified by the e-NeoSickle^®^ algorithm. Furthermore, 7.5% of the nonstandard profiles in Lille and 4.6% of the nonstandard profiles Dijon were misclassified, and 8% of the nonstandard profiles in Lille and 14.2% of the nonstandard profiles in Dijon were uninterpretable.

It is important to note that the nonstandard profiles accounted for less than 14% of the analyses in Dijon and 7% of the analyses in Lille. Overall, only 2% of the profiles in Dijon and 0.6% of the profiles in Lille were uninterpretable, and 0.9% of the profiles in Dijon and 0.6% of the profiles in Lille were misclassified.

The frequencies of “nonregular baseline” (cause 1) and “low/broad peak” (cause 2) features were similar at the two MS facilities. Our analysis showed that most of the nonstandard spectra in Dijon (54%) were due to a lack of resolution (cause 3), whereas this was the case for only 13% in Lille. Proportionally, the corrected phenotype FS (*n* = 71) was more likely to be labelled as “low resolution”, with respectively 31 and 36 abnormal spectra in Lille and Dijon. However, this low resolution had very little impact on the automatic classification, since less than 1% of all spectra labelled as “low resolution” were misclassified.

Sixteen percent of the spectra with a low, broad peak at 15,837 m/z were misclassified by the algorithm (Table 4). Of the 21 spectra of this type misclassified in Lille, 20 (95%) came from newborns with a corrected FAS phenotype. Likewise, of the 20 spectra of this type misclassified in Dijon (Table 4), 18 (90%) came from newborns with a corrected FAS phenotype. These spectra were very difficult to classify because the ratio between the β^S^ and β^A^ peaks was abnormal. A very low β^S^ peak (relative to the β^A^ peak) can be interpreted as a sample with high background noise or as a sample from an S homozygote or an AS heterozygote having received a blood transfusion (which induces a strong imbalance). Given that these spectra have low peak intensities, we recommend repeating the analysis or depositing a new sample. 

#### 3.2.5. Causes of Abnormal Spectral Features in Nonstandard MS Profiles

A spectral analysis showed that nonstandard profiles mainly resulted from the compilation of differing spectra within a quadruplicate. Figure 4 shows the variability in the raw data and its impact on data processing. The spectra in Figure 4a,b,e,f correspond to the raw data generated for the first (Figure 4a,b) and third (Figure 4e,f) of the quadruplicate sample depositions. The two depositions varied markedly in terms of the raw spectral intensity (less than 40 ua for deposition 1, and 200 ua for deposition 3). The data processing steps (including peak alignment and baseline normalisation and subtraction) normalise the spectrum intensity but also remove differences (see Figure 4c,g, corresponding to the normalized spectra, and a blow-up view of the zone of interest Figure 4d,h). The compilation of normalized data introduces a bias, and might lead to misinterpretation of the results by the software. Figure 4i shows a median spectrum with an irregular baseline; this “pseudo β^S^ peak” can be confused with a true β^S^ peak. However, the relative intensity of the pseudo β^S^ peak differed from that of a true β^A^ peak.

The analysis of each individual spectrum enabled us to affirm that at least one good-quality, high-resolution spectrum was obtained for each biological sample. The sample deposition method for MALDI-MS is known not to be highly reproducible, and thus makes it difficult to obtain very homogeneous results. Even though our study highlighted this difficulty, our analysis also showed that a high-quality result can be obtained systematically.

## 4. Occasional Misclassifications

The analysis of samples that were classified differently by the two analytical facilities confirmed that MALDI-MS can provide an interpretable result for each sample. In fact, the data in Table 5 showed that the 68 samples with a corrected FA phenotype that could not be interpreted in Dijon were correctly classified in Lille (67 standard spectra and 1 low-resolution spectrum). Likewise, the 21 samples with a corrected FA phenotype and that could not be interpreted in Lille were correctly classified at Dijon. Identical results were obtained with the corrected FAS samples. 

When a profile was misclassified or uninterpretable at one MS facility, the corresponding profile at the other MS facility was correctly classified and was less likely to be a nonstandard profile. Only one sample with a nonstandard profile was misclassified at both MS facilities (data not shown). A good classification with a standard spectrum was obtained for each sample on at least one of the MS facilities, which indicates that a low HbA concentration was not the main cause of misclassifications or uninterpretable results. Misclassifications or uninterpretable results were mainly due to the absence of a high-quality spectrum induced by technical incidents (tip blockage and non-deposition of the sample). Consequently, repeating the analysis of a sample with a nonstandard profile was the best way of obtaining a standard profile and a correct classification. Different data, such as peak intensities or the ratio between different peaks of β^A^ and β^S^ chains, allow for us to detect the nonstandard profile and to add an alert to the results.

## 5. Discussion

Newborn disease screening programmes are front-line public health measures, and as such must be based on robust analytical methods and data-processing software. Cost effectiveness is a further requirement, prompting the implementation of high-throughput screening units that reduce unit costs. Lastly, the maximal use of automation enables the analytical results to be validated with as little intervention as possible by the medical team. Our MALDI-MS platform and the associated data-processing and interpretation software were designed to address these challenges.

In newborn SCD screening with a MALDI MS system, the user-friendliness of the software interface and a high-throughput analysis coupled to automatic sample classification and traceability are directly related to practicability. As mentioned above, newborn SCD screening in France is centralised at a few specialist analytical centres. This centralisation requires a throughput of up to 600–1000 samples per day. The MALDI-TOF MS approach meets this requirement.

The next performance criterion of note concerns the method’s robustness: in other words, its ability to generate good-quality standard MS profiles that enable correct classification in agreement with the sample’s validated phenotype, whatever the newborn’s clinical status. In order to test our MALDI-MS under extreme conditions, we increased the proportion of difficult samples in the presently analyzed cohort; this corresponds to pathological specimens and specimens collected and/or stored under non-optimal conditions (samples from premature newborns, samples collected long after delivery, samples collected from France’s overseas regions and dependencies that may have been stored for a long time in a tropical atmosphere before delivery to the screening laboratory in continental France, etc.). In the present study, only samples from newborns having received one or more blood transfusions were discarded. The impact of blood transfusion will be described in a specific study of the effect of high HbA levels on the method’s sensitivity, specificity and resolution as a function of the β^s^ chain concentration. Moreover, we evaluated the MALDI-TOF method’s robustness by visually checking the spectral quality as well as by considering its ability to correctly classify the newborns as FA, FAS or FS. Benchmark MALDI-TOF MS spectra were thus established on the basis of the classification results and the visual quality of the spectra. The reference spectra correspond to all of the spectra defined as standard for which the information obtained is considered to be sufficient and of good quality. Indeed, visual spectral validation has proved to be most effective for revealing shot-to-shot variations.

Using this strategy, we determined the percentages of newborns with a standard profile and who were correctly classified (relative to their validated phenotype). The highest level of efficiency was obtained in the set of newborns with a corrected FA phenotype; around 85% of these analyses could be validated directly. A similar percentage was obtained for newborns with a corrected FAS phenotype. Moreover, nonstandard MS spectra were not systematically misclassified, and most were correctly classified. We consider these results to be very promising because it should be possible to further improve the procedures for sample deposition, raw data analysis and selection of the mass spectra to which the classification algorithm is applied.

None of the samples from FS newborns were misclassified, and a sample with an FAS-corrected phenotype was misclassified as an FA sample. This type of misclassification can lead to false negatives for newborns with an FSC phenotype. This major error required us to set up a strategy for correcting the automatic classification. The development and characteristics of tracking alerts for the nonstandard MS profiles will be described elsewhere. This approach should enable an occasional lack of reproducibility, sensitivity or resolution in Hb profiling to be detected.

There were two main reasons for the incorrect automatic classification of newborn phenotypes. Firstly, a low signal intensity at 15,850 and/or 15,880 *m*/*z* (i.e., similar to background noise) was intensified by the data processing (normalisation and alignment). Secondly, poor resolution of the ß and/or β^S^ chains generated signal overlap. The quadruplicate sample analysis improved the results, but was not always able to counterbalance shot-to-shot variability.

In a previous article [5], very poor spectral quality prevented the analysis of 20 of 844 samples (2.5%), even after several repeat analyses. The relatively high frequency of this incident was probably due to the inclusion of a high proportion of samples from premature newborns and/or HbS carriers. We have now resolved this problem. In most cases, a single repeat analysis was enough to obtain a standard profile. Some problems arose from time to time but were mainly related to robotic preparation of the blood spot samples. Further optimization of sample processing and the MS acquisition protocol should further improve the standardization of the MS profiles and reduce the frequency of repeat analysis.

Taken as a whole, our results for pooled standard and nonstandard spectra demonstrated NeoSickle^®^’s ability to classify correctly 97% of the samples tested in Lille and 98.8% of the samples tested in Dijon. Furthermore, only 2% of the nonstandard spectra in Dijon were uninterpretable, and less than 0.9% were misclassified. In Lille, 0.6% of the nonstandard spectra were uninterpretable and 0.6% were misclassified. It is important to note that these percentages were obtained with “difficult” blood samples (premature newborns, very late screening).

Our pilot study showed that the NeoSickle^®^ approach can differentiate between heterozygous FSE, FSO-Arab and S-β + samples on one hand and heterozygous FAS and homozygous FS samples on the other. Indeed, the signal intensity of the β^E^, β^O-Arab^ variants was much weaker than that of the β chain. In addition to classifying samples as FA, FAS or FS, studies of a larger cohort of patients might make it possible to clearly differentiate a fourth class (corresponding to heterozygous SX samples in which the β^X^ chain is E, O-Arab, a).

In conclusion, our new MALDI-TOF MS approach already meets today’s requirements [12] for large-scale, cost-effective newborn SCD screening, and is well-positioned to address future requirements for even greater throughputs and total automation to detect the HbS variants.

## Figures and Tables

**Figure 1 IJNS-05-00010-f001:**
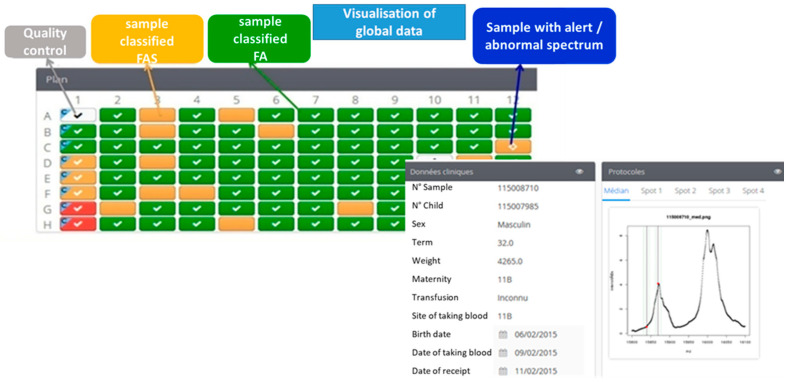
A screen display from the Sickle Cell Anemia Collect & Compare (SCACC) web application. Each box corresponds to a position of a sample on the 96-well plate. The green box corresponds to one sample classified FA, the orange box corresponds to one sample classified FAS and the red boxes are samples detected as FS. The software presents on the same page the clinical data (N° of sample, sex, term, weight, name of maternity, transfusion, location and date of taking blood, birth date) and the spectra of one sample.

**Figure 2 IJNS-05-00010-f002:**
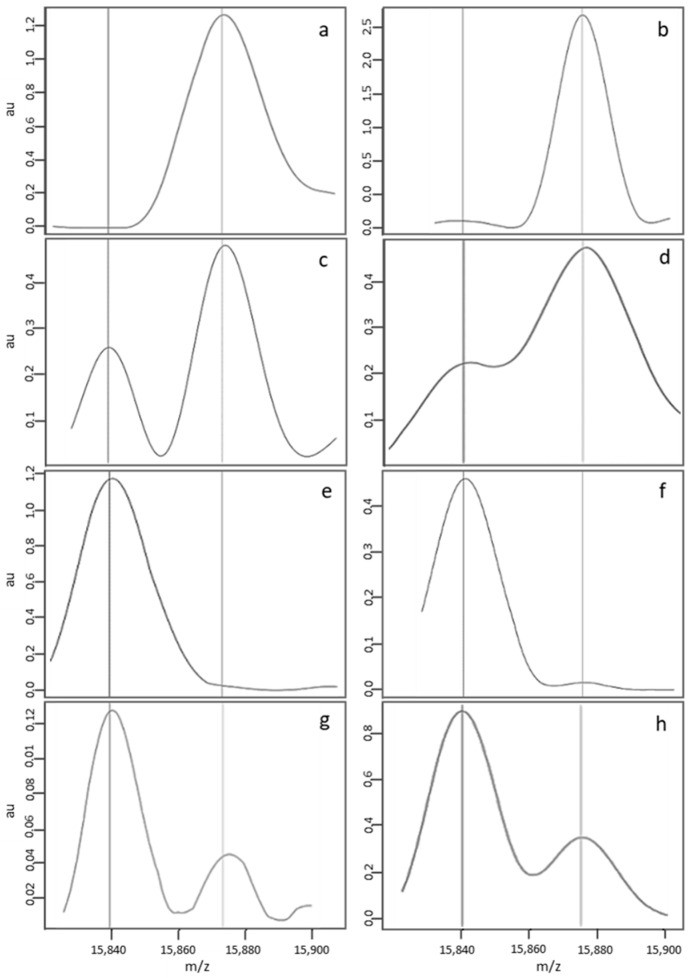
Examples of standard MS profiles correctly classified by the NeoSickle^®^ software, from newborns with the following phenotypes: FA (**a**,**b**), FAS (**c**,**d**), FS (**e**,**f**) and S-β + (**g**,**h**).

**Figure 3 IJNS-05-00010-f003:**
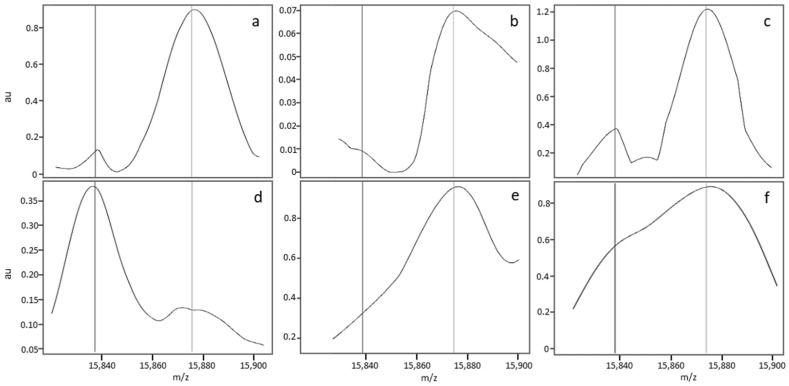
Zoom of MS profiles on the zone of interest for nonstandard FA (**a**,**b**,**e**), FAS (**c**,**f**) and FS (**d**) samples.

**Figure 4 IJNS-05-00010-f004:**
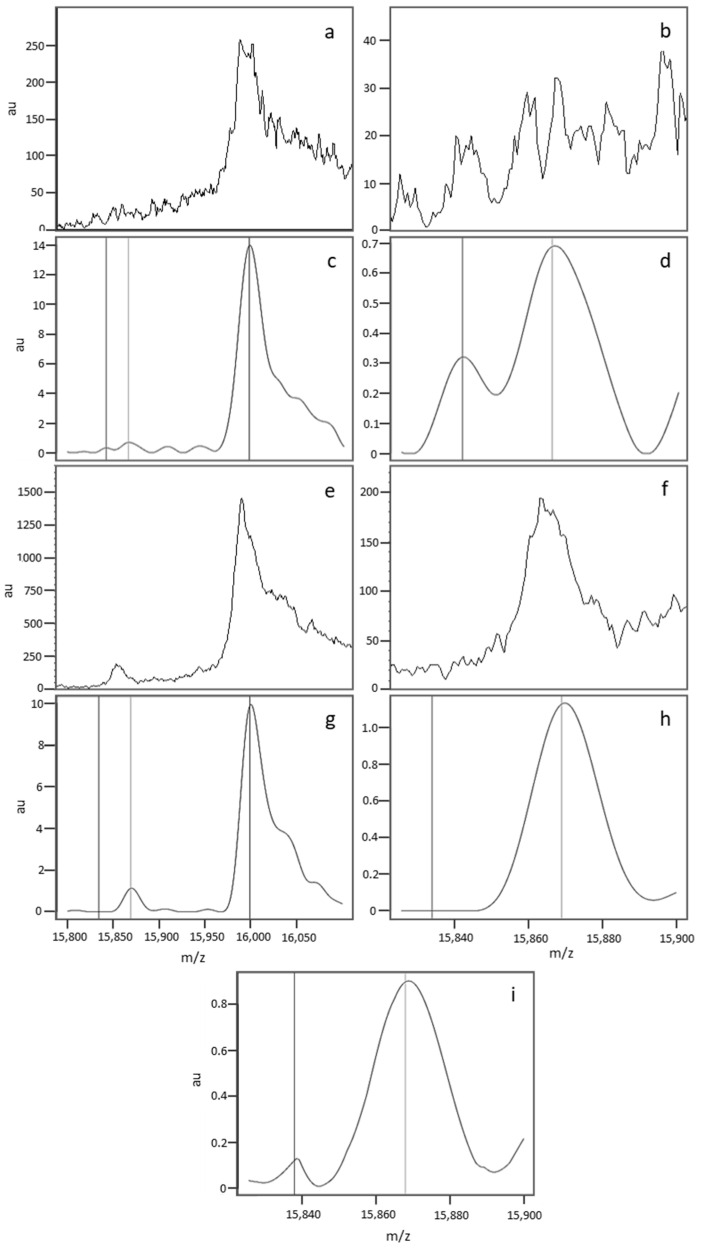
The spectra in (**a**,**b**,**e**,**f**) correspond to the raw data generated for the first (**a**,**b**) and third (**e**,**f**) of the quadruplicate depositions from the same sample. (**a**,**c**,**e**,**g**) show the whole spectrum (15,800–16,100 Da), and (**b**–**f**,**h**,**i**) show the region of interest (15,820–15,900 Da). The spectra in (**c**–**h** correspond to the processed spectra after alignment, baseline subtraction, and normalisation. The median spectrum (**i**) was obtained by compiling the individual normalized spectra.

**Table 1 IJNS-05-00010-t001:** Characteristics of the newborns and samples.

	Characteristics of the Neonates and Samples							
Phenotype	Number	Term			Sample Collection Date			
		23–32 weeks	33–36 weeks	>37 weeks	3–5 day	6–10 day	11–30 day	>30 day
FS	71	0	5	66	67	3	1	0
F/A	2834	203	809	1822	2651	90	77	16
F/AC	576	13	50	513	547	20	2	7
F/AE	141	2	9	130	138	2	0	1
F/AO-Arab	21	0	2	19	19	2	0	0
F/AD	17	0	0	17	17	0	0	0
F/AKorle-Bu	10	0	0	10	10	0	0	0
F/AX	86	4	3	79	81	3	0	2
F/C	7	0	1	6	6	0	0	1
F/E	3	0	0	3	2	1	0	0
F/O-Arab	1	0	0	1	1	0	0	0
Corrected FA	3696	222	874	2600	3472	118	79	
F/AS	2894	57	247	2590	2763	89	23	19
F/SC	23	0	1	22	22	1	0	0
F/SE	1	0	0	1	1	0	0	0
F/SO-Arab	1	0	1	0	1	0	0	0
Corrected FAS	2919	57	249	2613	2787	90	23	19
S-β^+^-thalassaemia	15	0	1	14	15	0	0	0

The number of corrected FA phenotype corresponding to the sum of all samples (FA, FAC, FAE, FAO-Arab, FAD, FAKorle-Bu, FAX, FC, FE, FO Arab where either β^A^ gene was duplicated [β^A^β^A^] or associated with other β chain variants (β^A^β^C^, β^A^β^E^, β^A^β^O-Arab^, β^A^β^D^, β^A^β^C^, β^A^β^X^) or where duplicate ß chain variants were found (β^C^β^C^, β^E^β^E^, β^O Arab^β^O-Arab^) that are detected FA by MALDI. The number of corrected FAS phenotype corresponding to the sum of all heterozygous (FAS) and composite heterozygous (FSC, FSE, FSO-Arab) samples where βS gene was associated respectively to βA gene or to other ß chain variants (β^S^β^C^, β^S^β^E^, β^S^β^O-Arab^) that are detected FAS by MALDI.

**Table 2 IJNS-05-00010-t002:** The number of standard MS profiles correctly classified (+) or misclassified (−), according to the newborn’s corrected phenotype.

	Lille MS Facility						Dijon MS Facility					
Number of Standard Profiles	**6181**						**5698**					
Corrected Phenotype	**FA**		**FAS**		**FS**		**FA**		**FAS**		**FS**	
Number of samples	3386		2753		37		3161		2507		35	
classification by the algorithm	+	−	+	−	+	−	+	−	+	−	+	−
	3386	0	2751	2	37	0	3161	0	2490	17	35	0

**Table 3 IJNS-05-00010-t003:** Classification by the algorithm of profiles considered to be nonstandard in a visual assessment. Some spectra could not be interpreted by the algorithm (IN). The percentage is based on the number of profiles considered to be nonstandard (780 profiles at Lille and 985 at Dijon).

			Corrected Phenotypes		
			FA	FAS	FS
MALDI-TOF MS classification of non-standard profiles	Lille	FA	282 (54.7%)	30 (5.9%)	0 (0%)
		FAS	7 (1.4%)	118 (23.2%)	0 (0%)
		FS	0 (0%)	1 (0.2%)	34 (6.7%)
		IN	21 (4.5%)	17 (3.4%)	0 (0%)
	Dijon	FA	446 (45%)	22 (2.2%)	0 (0%)
		FAS	21 (2.2%)	323 (32.7%)	1 (0.2%)
		FS	0 (0%)	1 (0.2%)	36 (3.7%)
		IN	68 (7.2%)	67 (6.8%)	0 (0%)

**Table 4 IJNS-05-00010-t004:** (A) Numbers and proportions of nonstandard spectra by corrected phenotype and by cause. (B) Numbers and proportions of correctly classified and misclassified spectra by cause. Cause 1 resulted in spectra with an irregular baseline, cause 2 resulted in spectra with a low, broad peak and cause 3 resulted in spectra with a low resolution. The percentage is based on the number of profiles considered to be nonstandard (780 profiles at Lille and 985 at Dijon).

A		Corrected Phenoztype			B		Correctly Classified	Misclassified
		FA	FAS	FS				
Lille	Cause 1	179 (37%)	90 (19%)	3 (0%)	Lille	Cause 1	252 (94%)	17 (6%)
	Cause 2	98 (21%)	28 (5%)	0 (0%)		Cause 2	105 (84%)	21 (16%)
	Cause 3	12 (3%)	31 (7%)	31 (7%)		Cause 3	77 (100%)	0 (0%)
Dijon	Cause 1	193 (23%)	69 (8%)	0 (0%)	Dijon	Cause 1	248 (95%)	14 (5%)
	Cause 2	103 (12%)	24 (3%)	0 (0%)		Cause 2	107 (84%)	20 (16%)
	Cause 3	171 (20%)	253 (30%)	36 (4%)		Cause 3	449 (98%)	11 (2%)

**Table 5 IJNS-05-00010-t005:** Paired classification of the samples by the two analytical facilities. Relationship with the spectral quality. This table summarizes the number of profiles correctly classified, misclassified and uninterpretable for each category of profile quality and for the two analytical laboratories.

Profile Abnormalities at 15,850 ± 10 *m*/*z*											
**Neonates with a Corrected FA Phenotype**											
**Automatic Classification of Matched Profiles**		**Irregular Base Line**		**Low, Broad Peak**		**Low Resolution**		**Regular Base Line**		**Uninterpretable**	
**Lille**	**Dijon**	**Lille**	**Dijon**	**Lille**	**Dijon**	**Lille**	**Dijon**	**Lille**	**Dijon**	**Lille**	**Dijon**
correctly classified	uninterpretable	1	0	0	0	0	0	67	0	0	68
correctly classified	Misclassified	4	10	7	0	1	5	17	0	0	0
uninterpretable	correctly classified	0	0	0	0	0	0	0	21	0	0
Misclassified	correctly classified	4	0	3	0	0	0	0	7	0	0
**Neonates with a Corrected FAS Phenotype**											
**Automatic Classification of Matched Profiles**		**Irregular Base Line**		**Low, Broad Peak**		**Low Resolution**		**Regular Base Line**		**Uninterpretable**	
**Lille**	**Dijon**	**Lille**	**Dijon**	**Lille**	**Dijon**	**Lille**	**Dijon**	**Lille**	**Dijon**	**Lille**	**Dijon**
correctly classified	uninterpretable	0	0	0	0	0	0	68	0	0	68
correctly classified	Misclassified	0	4	0	14	1	5	23	0	0	0
uninterpretable	correctly classified	0	0	0	0	0	1	0	16	17	0
Misclassified	correctly classified	12	0	12	0	0	4	6	26	0	0

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
