# Peer review of "A Multicentre Pilot Study of a Two-Tier Newborn Sickle Cell Disease Screening Procedure with a First Tier Based on a Fully Automated MALDI-TOF MS Platform"

_2409-515X, 2019, doi:10.3390/ijns5010010_

Round 1
Reviewer 1 Report
This study describes the MALDI-TOF and NeoSickle® software for the first stage of large-scale newborn screening for hemoglobinopathies are described in this study, which is nicely designed to build upon their previous experience as described in reference #5. They tested up 4 categories of blood samples: beta-globin variants, premature newborns, infants screened late, and routine activity. sickle cell disease, sickle trait, normal full-term, normal premature babies. The performance characteristics of two French reference labs using the MALDI-TOF-MS and NeoSickle® kit with these actual specimens from Guthrie cards are documented.
The study design is appropriate, with suitable scale that simulates the activity of a national newborn screening program. Comparison of the two labs was useful in illustrating the technical challenges. The level of technical detail is appropriate. The figures are clear. The language is clear.
Minor suggestions:
A. They use the SCACC web application for data handling, and presents results are presented in the form of “correctly classified”, “misclassified”, and “indeterminate”, and summarized as “98.8% correct classification.” The 98.8% depends on the samples selected and the frequency of these hemoglobins in France. This information might be presented more usefully for a newborn screening by adding calculation of the “sensitivity” and “specificity” in these categories: FAS, FS, FSA, FSX (compound heterozygotes where “X” is E or O-Arab, lines 204-210), perhaps suspected beta thalassemia, and then other beta-globin variants. The sensitivity and specificity would provide perspective at a level that a non-technical administrator or biologist for newborn screening could use to understand the high performance of the MALDI system for detection of HbS but limitations for some other beta globin variants (Lines 172-176). It would also allow ready calculation of positive predictive value and negative predictive value when combined with the expected frequencies of these beta globin variants in a population to be screened.
B. The summary in the abstract and the conclusion emphasize the “fully-automated” system. Limitations to complete automation of the MALDI-TOF technique are presented well in the body of the text – is there space to mention these limitations in the abstract and conclusions?
(1) cannot detect the difference between HbA and HbO-Arab, HbKorle-Bu and some unidentified variants (HbX). They designated these as “corrected HbA” samples. They plan to adjust their system to accommodate these and will devote a future study to these Hb variants.
(2) misclassification as FS by MALDI-MS came from composite heterozygous newborns classified as FS-O Arab and FSE by CE and HPLC. These required visual analysis by a biologist (Lines 204-210)
Author Response
This study describes the MALDI-TOF and NeoSickle® software for the first stage of large-scale newborn screening for hemoglobinopathies are described in this study, which is nicely designed to build upon their previous experience as described in reference #5. They tested up 4 categories of blood samples: beta-globin variants, premature newborns, infants screened late, and routine activity. sickle cell disease, sickle trait, normal full-term, normal premature babies. The performance characteristics of two French reference labs using the MALDI-TOF-MS and NeoSickle® kit with these actual specimens from Guthrie cards are documented.
The study design is appropriate, with suitable scale that simulates the activity of a national newborn screening program. Comparison of the two labs was useful in illustrating the technical challenges. The level of technical detail is appropriate. The figures are clear. The language is clear.
Minor suggestions:
A. They use the SCACC web application for data handling, and presents results are presented in the form of “correctly classified”, “misclassified”, and “indeterminate”, and summarized as “98.8% correct classification.” The 98.8% depends on the samples selected and the frequency of these hemoglobins in France.
Yes, we agree that the percentage depends on the samples selected, which is why we voluntarily presented in Table 1 the characteristics of all the samples in the study. The frequency of each phenotype in this study does not correspond to the frequency in France because the cohort was enriched in samples qualified as difficult (premature newborns, FAS samples, late and very late samples...). Some information have been included in the abstract to precise the characteristics of samples selected (22-24).
This information might be presented more usefully for a newborn screening by adding calculation of the “sensitivity” and “specificity” in these categories: FAS, FS, FSA, FSX (compound heterozygotes where “X” is E or O-Arab, lines 204-210), perhaps suspected beta thalassemia, and then other beta-globin variants.
The sensitivity and specificity would provide perspective at a level that a non-technical administrator or biologist for newborn screening could use to understand the high performance of the MALDI system for detection of HbS but limitations for some other beta globin variants (Lines 172-176).
Some information concerning of the “sensitivity” and “specificity” in the categories FA, FS and samples with HbS was included on the results (182-187).
For the other categories, it is not possible to perform a study of sensitivity and specificity because we have at the beginning done the postulate that the other variants are FA or FAS or FS "corrected". To avoid confusion we have modified lines 204-210.
It would also allow ready calculation of positive predictive value and negative predictive value when combined with the expected frequencies of these beta globin variants in a population to be screened.
The positive and negative predictive value for variant FS was included in text (188-192).
B. The summary in the abstract and the conclusion emphasize the “fully-automated” system. Limitations to complete automation of the MALDI-TOF technique are presented well in the body of the text – is there space to mention these limitations in the abstract and conclusions?
(1) cannot detect the difference between HbA and HbO-Arab, HbKorle-Bu and some unidentified variants (HbX). They designated these as “corrected HbA” samples. They plan to adjust their system to accommodate these and will devote a future study to these Hb variants.
(2) misclassification as FS by MALDI-MS came from composite heterozygous newborns classified as FS-O Arab and FSE by CE and HPLC. These required visual analysis by a biologist (Lines 204-210)
We added the notion of detection of HbS linked at the notion of “Full automated” procedures. It is clear that all samples containing HbS variant need a visual validation and a test of validation.
Reviewer 2 Report
very interesting scientific work, in order for more easy reading text need revision and shortening/tightening, where the aim of detecting SCD is the main goal in the context of NBS.
All tables and figures and tables should be with explaining texts in the heading or subtext. Axis should be named with measure/indicator and unit, either in text or figure.
Paragraph: Material and methods needs revision:
"routine activity for certain days", "normal CE" is not clear, sample and cohort properties and preanalytic handling should be worked out more clearly (what is “clinical data”, “validated results using a reference method”?),
sample preparation is insufficiently described, as Neo sickle® kit application is not described: at least for example: used according to the manufacturers instructions, in short… (solution 1, solution 2 , water, wells, volume,…. ) or as published in (reference) in short …
analytical data flow: reference? Or data (which parameters fed the algorithm? Or how the phenotyped sample have been phenotyped?) providing evidence for these statements. What is a “median newborn profile”?
table 1 needs revision, it would be more clearly to know were a sum is presented of which phenotype, and that in the so called corrected FAS there are sickle cell disease phenotypes included,
the term "corrected phenotype” may be misleading, it should be more descripting like "MS suitable phenotype” and the definition comes rather late in the text in the results part.
3.3.1: what is “normal” (fig 2f) does it mean FA?
3.3.3: peaks should be named consistent (beta, beta S, beta A)
if all sample have been measured at both facilities why different numbers (6,181/5,698)?
The tables 2,3 and 4 seem somehow contradictive and numbers and percentages of standard and non-standard spectra are somehow mixed in the text (results and conclusion) and should more clarified. Percentages given in table 3 and 4 may be more confusing as it is not clearly stated what namber they refer to (to the whole set of samples, only non-standard profiles etc). If a ratio is used a ration should be explained (as the spactra consist already on ratios, additional ratios remain unclear (3.3.4)
It should be stated how the workflow/algorithm is, depending on the spectra to decide on further analysis, second sample, second tier method,… This algorithm may be underlined by absolute and relative frequencies found in this study. If a recommendation of repeating a sample is given, was it successful in the study course or not applicable for other reason? The aim to correctly identify children with SCD should not be hided behind the table 5. Table 5 may be a supplement. Terms as classified, mis- classified, interpretable and uninterpretable used abundantly may implicate a more problematic view which seem not the case regarding your results (chapter 4). A more clear presentation could help.
Discussion: critical points: Visual checking versus algorithm of the software? The term “benchmark MS spectra” seem not introduced? The misclassification of FAS for FA (of standard profile) should be more extensively discussed how to avoid, as it not only may be false negative for FSC but other compound heterozygous states of sickle cell disease. Please explain how MALDI-TOF MS can be implemented in the screening process in France (or other countries) from now, before concluding it does meet all requirements already. A prospective study approach should be considered after this (blinded?) retrospective approach.
Author Response
Comments and Suggestions for Authors
very interesting scientific work, in order for more easy reading text need revision and shortening/tightening, where the aim of detecting SCD is the main goal in the context of NBS.
All tables and figures and tables should be with explaining texts in the heading or subtext. Axis should be named with measure/indicator and unit, either in text or figure.
The explanation texts was added in subtext of each table and figures
Paragraph: Material and methods needs revision:
"routine activity for certain days", "normal CE" is not clear, sample and cohort properties and preanalytic handling should be worked out more clearly (what is “clinical data”, “validated results using a reference method”?),
This information have been added to the text.
sample preparation is insufficiently described, as Neo sickle® kit application is not described: at least for example: used according to the manufacturers instructions, in short… (solution 1, solution 2 , water, wells, volume,…. ) or as published in (reference) in short …
This information have been added to the text.
analytical data flow: reference? Or data (which parameters fed the algorithm? Or how the phenotyped sample have been phenotyped?) providing evidence for these statements. What is a “median newborn profile”?
This information have been added to the text.
table 1 needs revision, it would be more clearly to know were a sum is presented of which phenotype, and that in the so called corrected FAS there are sickle cell disease phenotypes included,
the term "corrected phenotype” may be misleading, it should be more descripting like "MS suitable phenotype” and the definition comes rather late in the text in the results part.
Some changes have been made to Table 1 to make it more visible.
The reviewer 2 indicates that the sum in the FAS includes SCD phenotypes, the reviewer obviously targets the FSC which is normal. But as indicated the objective of this first line screening is to detect HbS in all samples. Since all samples containing HbS must undergo a confirmation step, SCD phenotype type FCS will then be detected and validated during the validation phase using the other method like HPLC or CE .
We have reintroduced the paragraph "Definition of corrected phenotypes" in the material and method to meet the request of the reviewer 2.
3.3.1: what is “normal” (fig 2f) does it mean FA?
The text has been modified to avoid confusion
3.3.3: peaks should be named consistent (beta, beta S, beta A)
The text has been modified to meet the request of the reviewer
if all sample have been measured at both facilities why different numbers (6,181/5,698)?
This is explained by the fact that all the spectra are not classified as standard on both MS facilities as indicated in lines 179-180, the percentage indicated on line 179-180 applied to the number of samples makes it possible to find the number of the table 2
The tables 2,3 and 4 seem somehow contradictive and numbers and percentages of standard and non-standard spectra are somehow mixed in the text (results and conclusion) and should more clarified.
I agree with the reader's approach and have corrected the errors
Percentages given in table 3 and 4 may be more confusing as it is not clearly stated what namber they refer to (to the whole set of samples, only non-standard profiles etc).
The explanation texts were added in subtext of each table 3 and 4
If a ratio is used a ration should be explained (as the spactra consist already on ratios, additional ratios remain unclear (3.3.4)
Clarifications have been included to better understand the calculations made
It should be stated how the workflow/algorithm is, depending on the spectra to decide on further analysis, second sample, second tier method,… This algorithm may be underlined by absolute and relative frequencies found in this study. If a recommendation of repeating a sample is given, was it successful in the study course or not applicable for other reason? The aim to correctly identify children with SCD should not be hided behind the table 5. Table 5 may be a supplement. Terms as classified, mis- classified, interpretable and uninterpretable used abundantly may implicate a more problematic view which seem not the case regarding your results (chapter 4). A more clear presentation could help.
Clarifications have been included and the text has been modified to have a better reading of the document
Discussion: critical points: Visual checking versus algorithm of the software? The term “benchmark MS spectra” seem not introduced? The misclassification of FAS for FA (of standard profile) should be more extensively discussed how to avoid, as it not only may be false negative for FSC but other compound heterozygous states of sickle cell disease. Please explain how MALDI-TOF MS can be implemented in the screening process in France (or other countries) from now, before concluding it does meet all requirements already. A prospective study approach should be considered after this (blinded?) retrospective approach.
This study shows the potential use of the MALDI-MS approach for the detection of HbS and indicates the necessary alert points to put in place to consolidate the results and avoid any misclassifications. The results obtained by the algorithm were compared to a visual analysis of spectra and phenotypes (validated by CE and HPLC) in order to measure the ability of the algorithm to classify samples or to give a doubt alert. This first algorithm set up for this study shows its performance although it is obvious that improvement points need to be added. Some spectral data have been used to qualify the spectra as "standard" in order to evaluate the ability of the algorithm to achieve self-validation. The term "MS spectra benchmark" corresponds to the spectra defined as standard for which the quality of the information produced is considered sufficient.
The misclassification of FAS samples into FA (of standard profile) by MALDI is indeed a critical point, which will require optimization of the parameters. As indicated they were spectra whose overall intensity was low due to poor quality MALDI target deposition. This global spectral intensity parameter will therefore be introduced in the next version of the algorithm. The analysis of the data of overall intensity of the spectrum, intensity of the peaks of interest and resolution will make it possible to select the spectra whose quality will ensure a negative predictive value of presence of HbS of 100%. Spectra that do not meet these criteria will be considered non-standard and samples / or deposits will have to be re-analyzed to obtain a "standard" spectrum.
As indicated on line 369-370 (uncorrected version) to ensure the quality of the results the development and characteristics of tracking alerts for the non-standard MS profiles has been realized and will be described elsewhere.
The recommendations from the Pan-European Consensus Conference indicate that the new methods currently being tested should to be as specific and sensitive as HPLC and CE before implementation on a larger scale. In this study we show that the detection of HbS chain in samples was obtained on the Lille and Dijon MS facilities with a sensitivity of 98.4% and 97% and with a specificity of 99.8% and 99% on the basis of samples for which a result was obtained in HPLC and CE. In addition, the compilation of data from Dijon and Lille shows that in the absence of a technical incident inducing poor quality spectra, the sensitivity and specificity for the detection of HbS is 100% on the samples that gave results. by HPLC or CE.
Moreover, other results show that the NeoSickle® approach combined with the MALDI-MS analysis gives results in 90% of the samples for which the CE could not give results. These data are not presented in this study because we wanted to focus on samples for which the classification result was validated by one or the other of the two reference methods (CE or HPLC).
Another study on a larger number of samples will validate the approach and implement the NeoSickle® approach in the screening process in France.